# Psilocybin-assisted therapy for the treatment of resistant major depressive disorder (PsiDeR): protocol for a randomised, placebo-controlled feasibility trial

James Rucker ![ORCID],[1,2] Hassan Jafari,[3] Tim Mantingh,[1] Catherine Bird,[1] Nadav Liam Modlin,[1] Gemma Knight,[1] Frederick Reinholdt,[1] Camilla Day,[1,2] Ben Carter ![ORCID],[3] Allan Young[1,2]

[1]Department of Psychological Medicine, King's College London, London, UK
[2]National Affective Disorders Service, South London and Maudsley NHS Foundation Trust, London, UK
[3]Department of Biostatistics, King's College London, London, UK

**Correspondence to**
Dr James Rucker;
james.rucker@kcl.ac.uk

## ABSTRACT

**Introduction** Psilocybin-assisted therapy may be a new treatment for major depressive disorder (MDD), with encouraging data from pilot trials. In this trial (short name: PsiDeR) we aimed to test the feasibility of a parallel-group, randomised, placebo-controlled design. The primary outcomes in this trial are measures of feasibility: recruitment rates, dropout rates and the variance of the primary outcome measure of depression.

**Methods and analysis** We are recruiting up to 60 participants at a single centre in London, UK who are unresponsive to, or intolerant of, at least two evidence-based treatments for MDD. Participants are randomised to receive a single dosing session of 25 mg psilocybin or a placebo. All participants receive a package of psychological therapy. The primary outcome measure for depression is the Montgomery Asberg Depression Rating Scale collected by blinded, independent raters. The primary endpoint is at 3 weeks, and the total follow-up is 6 weeks. With further informed consent, this study collects neuroimaging and omics data for mechanism and biomarker analyses and offers participants an open label extension consisting of a further, open label dose of 25 mg of psilocybin.

**Ethics and dissemination** All participants will be required to provide written informed consent. The trial has been authorised by the National Research Ethics Committee (20-LO/0206), Health Research Authority (252750) and Medicine's and Healthcare Products Regulatory Agency (CTA 14523/0284/001-0001) in the UK. Dissemination of results will occur via a peer-reviewed publication and other relevant media.

**Trial registration numbers** EUDRACT2018-003573-97; NCT04959253.

### Strengths and limitations of this study

► Allows a direct comparison of adverse event data between treatment groups.
► Allows a robust measurement of the variance in the primary outcome measure between groups to inform an efficacy trial.
► As this is a single-centre study, the findings are unlikely to be generalisable beyond the trial setting.
► While raters of the primary outcome are independent of the study team, they are not independent of the institution undertaking the research.
► Psilocybin has a subjectively noticeable effect, which may result in participants and clinical care team being able to guess allocation.

is estimated by the UK government to be £20.2–£23.8 billion per year.[2] Treatment resistant depression (TRD), defined as failure to respond to at least two antidepressants at therapeutic dose for at least 6 weeks, is seen in 1/3 of those who suffer from clinical depression, and as resistance to treatment develops, prognosis worsens.[3] Except for esketamine and transcranial magnetic stimulation, there have been no significant medical treatment developments for MDD since the introduction of selective serotonin reuptake inhibitors in the late 1980s.

The classical psychedelic drugs, including psilocybin and d-lysergic acid diethylamide, were marketed as treatments before classification in Schedule 1 of the Misuse of Drugs Regulations in 1971, which effectively ended routine clinical use and research.[4] Despite this classification, they were not considered dangerous when used in medically controlled settings.[5–7]

## INTRODUCTION

Major depressive disorder (MDD) is a disabling and economically costly mental health problem. Its prevalence in England is estimated to be between 29 and 42 per thousand people.[1] The estimated cost in England

A systematic review in 2016 of trials published prior to prohibition detailed trials of psychedelics in patients with broadly defined depression.[8] While pre-prohibition studies were often of suboptimal design, at high risk of bias and did not collect consistent outcome measures, in aggregate 335 of 423 (79.2%, median 80.0%) patients in 19 separate studies were judged by their clinicians to have at least 'improved' with psychedelic therapy.

Psilocybin was isolated and synthesised by Albert Hofmann in 1957 and 1958, respectively.[9] Psilocybin is a partial agonist at the 5-HT2A receptor in the brain, and this likely accounts for the characteristic subjective effects.[10] However, its activity at serotonin receptors is broad.[11] It has low or absent affinity for the serotonin transporter, dopaminergic, histaminergic, adrenergic or cholinergic receptors.[12] Psilocybin is not habit forming in either humans or animals.[13] The subjective effects include heightened emotions, mystical experiences, misperceptions and blurring of conceptual boundaries. Some users of psilocybin have reported benefits in mental well-being that continue for weeks or after the dose.[14]

In a resurgence of clinical interest, government licensed clinical trials using psilocybin in phase 1 and 2 designs have been published since 2000.[14–23] We completed a phase 1 randomised, placebo-controlled safety study in which 89 healthy volunteers were randomised 1:1:1 to receive a single dose of placebo, 10 mg of psilocybin or 25 mg psilocybin in a clinical research facility, with 3 months of clinical follow-up. In this trial, there were no serious adverse events (AEs), no AEs that led to withdrawal and data from cognitive tasks indicated no negative effects of psilocybin when compared with placebo.[24]

In an open label pilot study of oral psilocybin in adults with TRD, 20 patients were given two oral doses of psilocybin (10 mg and 25 mg) 1 week apart in a clinical research facility with psychological support before, during and after the experience.[14 17] Both the 10 mg and 25 mg doses were well tolerated; no serious or unexpected AEs occurred. One patient withdrew from the trial after treatment, citing lack of efficacy. Patient rated depression severity was markedly reduced in comparison to baseline at 1 week after 25 mg psilocybin (Cohen's d=2.2, p<0.001) and 5 weeks (Cohen's d=2.3, p<0.001), remaining positive at months 3 and 6 following treatment (Cohen's d=1.5, p<0.001 and d=1.4, p<0.001, respectively). However, this study was not blinded and did not have a placebo control group. Expectation and/or the psychological support provided as part of the trial may account for the improvements in depression scores. Furthermore, it could not be determined from the open label design whether a randomised, placebo-controlled design investigating psilocybin would be acceptable to patients with TRD.

Thus, the purpose of this trial is to evaluate the feasibility and safety of psilocybin-assisted therapy, given under supportive conditions, compared with placebo-assisted therapy in a randomised, blinded trial design in adult participants with treatment resistant MDD. A further purpose of this trial was to estimate the variance of the primary outcome measure between groups to inform a power calculation for a future efficacy trial.

Since the inception of this trial, other feasibility trials have been published with psilocybin in non-TRD patient samples.[25 26] However, as well as not focussing on a treatment resistant population, these trials did not include a true placebo control. In discussion with the funders of this trial, we decided to use a true placebo control to measure the difference in AE rate and the variance in the primary depression outcome measure without the difficulties implied by an active control.

## TRIAL OBJECTIVES

The primary objective is to evaluate the feasibility of a randomised, controlled trial design, in which a single dose of psilocybin 25 mg PO versus placebo, is given to adult participants with treatment resistant MDD (TRD), under psychologically supportive conditions, with 6 weeks of follow-up, by measuring recruitment rates, dropout rates and by estimating the variance of the primary outcome measure (Montgomery Asberg Depression Rating Scale (MADRS)) to inform on the design of a phase 3 trial.

The secondary objectives of this trial are as follows:

1. To assess the clinician-rated efficacy of psilocybin 25 mg compared with placebo via: (a) the MADRS total score during follow-up; (b) the proportion of participants who demonstrate a response to treatment, where response to treatment is defined as a ≥50% decrease in MADRS total score during follow-up; (c) the proportion of participants in remission, where remission is defined as a MADRS total score ≤10 at week 3.

2. To assess the participant-rated efficacy of psilocybin 25 mg compared with placebo via: (a) the proportion of participants who demonstrate a response to treatment, where response to treatment is defined as a ≥50% decrease in The Quick Inventory of Depressive Symptomatology (QIDS-SR-16) total score from baseline at week 3; (b) the proportion of participants in remission, where remission is defined as a QIDS-SR-16 total score ≤5 at week 3.

3. To evaluate the safety and tolerability of psilocybin in participants with TRD based on AEs, changes in vital signs and suicidal ideation/behaviour (measured using the Columbia Suicide Severity Rating Scale).

The primary outcomes in this trial are measures of feasibility: recruitment rates, dropout rates and the variance of the primary outcome measure of depression. The secondary and exploratory objectives of this trial are specified in online supplemental file 1.

### Trial design

The trial is a parallel group, two arm, double-blind, randomised, placebo-controlled, between-subjects, single centre, exploratory design in up to 60 participants. The trial is taking place between September 2020 and September 2023 in a single centre in London, UK. Because of the impact of the COVID-19 pandemic, the

recruitment period may be extended, subject to necessary approvals. All study visits and dosing sessions take place in the Clinical Research Facility of King's College Hospital, Denmark Hill, London, UK.

The trial is funded by the National Institute for Health Research Clinician Scientist Programme (CS-2017-17-007). The sponsor of this trial is King's College London and the South London and Maudsley NHS Foundation Trust. The trial protocol was peer-reviewed at grant application and by an internal process of risk, capability and capacity assessment by delegates on the sponsor.

## Patient

Three participants from a similar pilot trial[17] were engaged in the design of this trial, with one participant agreeing to implement a funded, participant-facing website (https://psider.info/). We presented this trial to the sponsor's Service User Research Enterprise programme during the design of the trial to seek further feedback. Subsequently, five people with lived experience of mental health difficulty were recruited to the trial steering committee and have reviewed participant facing written material and contributed to the overall design of the trial in addition to ongoing oversight of the trial.

Since the study is exploratory, formal sample size calculations were not undertaken. The sample size selected was informed by published work that models processes of parameter estimates for continuous measures from early phase trials.[27] We aim to recruit 60 participants, randomised 1:1 to treatment versus placebo arms. This is considered sufficient to test the intervention and provide adequately reliable estimates of recruitment rate and loss to follow-up. The expected rate of loss to follow-up is 20%. With 60 participants, we would be able to estimate this target rate with a 95% CI of ±10.1% (9.9% to 30.1%).

## Trial activities

A schematic of trial activities is provided in figure 1. The full schedule of events is detailed in online supplemental file 1.

Recruitment occurs via referrals from primary and secondary care, psychological therapy services, established clinical trial registries of patients who have consented to contact from study teams and directly from advertising in the community and on social media. All self-referrals are directed to seek a referral from their general practitioner or mental health professional to the study team.

## Screening

Participants expressing interest in the trial are directed to a brief online survey. This contains a link to the Participant Information Sheet and collects initial consent to store and process basic clinical and demographic data to determine initial eligibility criteria. Those who fulfil initial eligibility criteria for the study are further assessed by telephone or video call to collect further information.

Those who are deemed to be potentially eligible are invited to a face-to-face screening visit with a member of the study team. A study website is available that contains additional information for participants. Written informed consent is taken at the screening visit by a physician.

## Eligibility criteria

The main inclusion criteria for this trial are:
- ▶ 25–80 years of age.
- ▶ Fluency in English.
- ▶ Able to give informed consent.
- ▶ Fulfil Diagnostic and Statistical Manual of Mental Disorders (5th Edition) (DSM-5) criteria for a primary diagnosis of current single or recurrent episodes of MDD of at least moderate severity but without psychotic features as defined in the Mini-International Neuropsychiatric Interview (MINI V.7.0).
- ▶ A score of at least 14 on the Hamilton Depression Rating Scale for Depression (HAM-D).
- ▶ A score of at least 5 out of 15 on the Maudsley Staging Method, which defines the degree of treatment resistance in depression.[28]
- ▶ For those participants over the age of 60, the first episode of depression must have started prior to their 60th birthday.
- ▶ Participants must have failed to respond to two or more antidepressants prescribed at the minimum effective dose for at least 6 weeks, or at least one antidepressant prescribed at the minimum effective dose for at least 6 weeks AND a course of evidence-based psychotherapy given for at least six sessions. The definition of 'failed to respond' is an inadequate response to an adequate duration and dose, or failure to reach an adequate dose and duration

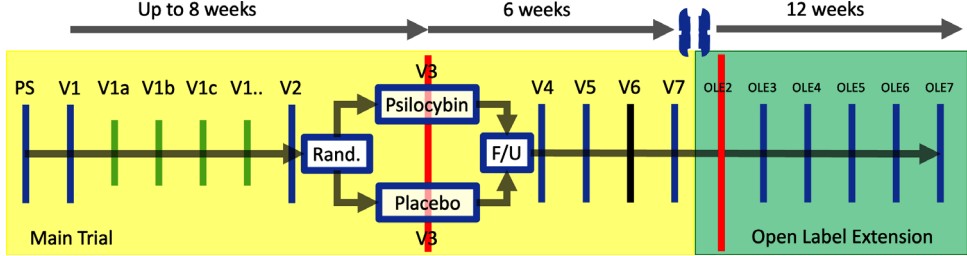

**Figure 1** Trial Schematic. Blue, red and black lines represent in-person visits. Green lines represent a flexible visit schedule of psychological preparation, which may be undertaken remotely. Red lines represent dosing sessions. Dosing session at OLE2 consists of 25 mg of psilocybin for all eligible and consenting participants. Black line represents the primary end point (3 weeks after V3). PS, prescreening.

due to lack of tolerance. We define 'evidence based psychotherapy' with reference to existing National Institute for Clinical Excellence guidelines for the treatment of depression.[29]

The main exclusion criteria for this trial are:

► DSM-5 diagnoses (ascertained by the MINI V.7.0) of bipolar affective disorder type 1 or type 2, any psychotic disorder (except psychosis occurring during acute intoxication with a substance), any drug or alcohol dependence syndrome, any personality disorder or any dementia. Diagnoses made by the MINI V.7.0 are subject to clinical confirmation by a psychiatrist.

► Suicide attempt in the past year requiring admission to hospital.

► Depression secondary to another medical condition.

► A medical diagnosis incompatible with psilocybin treatment, such as uncontrolled diabetes mellitus.

► Inability to provide a blood sample or ECG.

► Clinically significant biochemical or electrocardiographic abnormalities.

► Women who are pregnant, breast feeding or unable to use adequate contraception during the trial.

► Those who are not registered with a primary care doctor or who do not consent to sharing of information between the primary care doctor and the study team.

Full inclusion and exclusion criteria are detailed in online supplemental file 1.

### Preparation

If initial eligibility is confirmed, participants enter a preparatory phase of up to 8 weeks. The purpose of the preparatory phase is as follows:

1. Allow time for antidepressant medications to be tapered under medical supervision.
2. Enable participants to engage with a psychological therapist (minimum of 3 hours).
3. Allow gathering of collateral information to inform a final eligibility decision.

The purpose of the psychological therapy during the preparation visit is:

1. Building mutual trust and establishing processes of practical and interpersonal support for the dosing session.
2. Practicing a range of established techniques for the management of difficult emotional material.
3. Psychoeducation around the drug experience.

The baseline visit occurs at the end of the preparation phase. Final eligibility is confirmed by a physician. Baseline measures of the primary outcome and other measures are collected, including neuroimaging and biological sample acquisition, if consented to.

Participants are randomised 1:1 at the baseline visit to one of two study arms (treatment or placebo).

► Treatment consists of a single dosing session consisting of 25 mg of psilocybin given in a medically supervised and supportive setting.

► Placebo consists of a single dosing session consisting of placebo given in a medically supervised and supportive setting.

Randomisation uses a web-based service hosted at the sponsor's Clinical Trials Unit in accordance with a standard operating procedure and held on a secure server. The allocation sequence is generated using minimisation, with sex (male, female) age group (25–59, and 60+) and past psilocybin use (yes, no) as factors. The sequence is concealed from all researchers and the trial statistician. Allocation is performed by staff in the local hospital pharmacy. Strict allocation concealment is maintained.

### Dosing

The dosing visit takes place 1–3 days after the baseline session. The setting for dosing is a quiet, neutrally furnished room with a reclining seat or bed. The treatment and placebo consists of five capsules of 5 mg psilocybin or five capsules of inactive placebo (starch 1500), respectively. Psilocybin and placebo capsules are provided by Compass Pathways.

The intervention is taken by mouth with a glass of water. A mouth check is performed. Participants are then assisted to relax and encouraged to direct their focus internally. They are given an eye-mask and headphones with relaxing music. A member of the study team (usually the psychological therapist) is always with the participant. The role of the psychological therapist is to provide:

1. Practical support.
2. Breathing and relaxation techniques.
3. Encouraging an open curiosity around experiences as they arise.

Participants are medically assessed and discharged at the end of the dosing session, usually in the care of a friend or relative. Overnight stay can be arranged, if required.

Since the intervention consists of a single dose of a drug, there are no criteria for discontinuing the intervention or for monitoring adherence. Benzodiazepines and antipsychotics are available if pharmacological management is determined to be required by the study psychiatrist during a dosing visit.

### Follow-up

The follow-up period is 6 weeks. Face to face follow-up with the study team occurs at 1 day, 1 week, 3 weeks and 6 weeks after the dosing session.

As part of follow-up, a total of at least 4 hours of post dosing psychological support is provided to participants. The purpose of the psychological therapy during follow-up is integration, and consists of:

1. Normalisation of emotional content and experiences post dosing.
2. Emotional and experiential exploration of material relevant to depression.
3. Implementing insights into daily life and providing structure for further psychological work.

## Participant retention

Participants receive ongoing psychological therapy as part of this trial. An open label extension to this trial is described in which all eligible participants receive an open label dose of the intervention and a further period of psychological therapy. These interventions may improve retention.

## Outcomes
### Feasibility outcomes

### Recruitment capability

For evaluating the recruitment capabilities, we will report numbers of patients screened, not found eligible, refusal rate before randomisation, excluded due to safety concerns, and numbers found eligible and recruitment rate at baseline (final eligibility). The participation rate will be computed as the number of patients enrolled divided by the number approached to participate. The eligibility rate will be computed as the number of patients eligible divided by approached (screened). Randomisation rate will be computed as the number of patients randomised divided by eligible patients. We will also evaluate the feasibility of the eligibility criteria by reporting the number of people excluded according to the different exclusion criteria to understand whether the criteria were clear and sufficient or were too inclusive or restrictive.

### Follow-up and adherence to the allocated treatment

Follow-up rate will be reported as the number of participants at weeks 1, 3 and 6 follow-ups postrandomisation by treatment arm. The number and proportion of withdrawals from the trial and the reason for withdrawal will be summarised at each follow-up and by treatment arm.

The number and proportion of participants with missing data for each baseline, primary and secondary variables will be summarised as a whole and by group and follow-up time point. Descriptive statistics will be used to compare the baseline characteristics of the missing follow-ups (at 1, 3 and 6 weeks) with those of the complete follow-up and, if possible, based on the number of cases, using the logistic predictors of the missing model.

### Clinical outcomes

The primary outcome measure for depression is the MADRS, collected by raters independent of the study team but not the institution undertaking the research. The MADRS is a well validated scale of depressive symptoms known to be sensitive to change.[30]

The secondary outcomes are the QIDS-SR-16,[31] GAD-7 (Generalised Anxiety Disorder Assessment),[32] EQ-5D-5L (Euro-QOL Health Questionnaire),[33] Work and Social Adjustment Scale,[34] will be measured at baseline, 3 and 6 weeks.

HAM-D-17[35] and MINI V.7.0[36] will be collected only as baseline/screening measures.

## Blinding

Bottles are labelled by a code that does not reveal the identity of the contents. Each bottle contains five opaque capsules. Capsules of psilocybin and placebo are matched for weight, colour, taste, and smell.

Participants, research staff, clinical staff and the trial statistician are kept blind to allocation throughout the trial; however, psilocybin induces an altered state of consciousness that may allow subjects and researchers to predict allocation. To mitigate against bias in the collection of the primary outcome (MADRS), raters are independent of the immediate study team but not independent of the institution undertaking the trial. Raters are trained to collect the MADRS using the Structured Interview Guide for the MADRS[37] and required to ask participants not to discuss any information that might reveal their beliefs about treatment allocation. Participants are asked about their beliefs about allocation via an electronic scale.

In the event of a serious unsuspected serious adverse reaction in which pharmacovigilance authorities require knowledge of treatment allocation, unblinding may be undertaken by a representative of the sponsor without unblinding the study team. The blind may be broken in an emergency at any time by contacting the pharmacy.

## Trial optional components
### Biological sample acquisition

Optionally, participants may consent for the collection and storage of five sets of venous blood samples. The first set is collected at baseline, with four subsequent sets collected at 1 day, 1 week, 3 weeks and 6 weeks after dosing. The analysis of these samples is not an intended outcome of this study.

### Neuroimaging

Optionally, participants may consent to two 1-hour functional MRI scans. The first is taken at baseline and the second at 1 week after dosing. The neuroimaging paradigm includes resting state, an emotional faces paradigm and magnetic resonance spectroscopy. The analysis of this dataset is not an intended outcome of this study. The inclusion criteria for this element of the study are eligibility for the main study and absence of contraindications to an MRI scan.

### Open label extension

Optionally, all eligible participants at 6-week follow-up are offered an open label extension to the main trial. This is illustrated schematically in figure 1. The inclusion criteria for this element of the study are informed consent and the absence of the development of exclusion criteria as defined for the main trial. The open label extension comprises a further, single dosing session of 25 mg of psilocybin given in an identical setting, with follow-up at 1 day, 1 week, 3 weeks, 6 weeks and 12 weeks.

## End of trial

The end of the trial is defined as the timepoint of the database lock.

## Psychological therapist training

Psilocybin assisted therapy is a nascent field, and thus competencies for therapist training is in ongoing development. The research team undertaking this trial are recognised leaders in this field. All psychological therapists in this trial were professionally qualified and underwent a 3-day training course provided by NLM, GK, JR and overseen by FR.

## Trial monitoring

The trial is monitored for regulatory compliance and quality by the King's Health Partners Clinical Trials Office, on behalf of the sponsor. A trial steering committee meets two times a year to supervise the conduct of this trial on behalf of the trial sponsor and funder. An independent data monitoring and ethics committee is set up for this trial, which has access to the full data set and may make recommendations to the sponsor about continuing the trial based, as necessary. No interim analysis is planned, given that the primary aim of the trial is feasibility.

## Statistical analysis

Analysis of feasibility parameters will use descriptive statistics and confidence intervals where appropriate. Outcome variables will be summarised by trial arm (and standardised effect sizes) estimated for baseline, 3 and 6 weeks after randomisation. 95% CIs will be constructed for the trial arm differences.

Analysis of the primary outcome for depression will assess the treatment group difference at 3 weeks with fixed effects of treatment group in the model and will be adjusted for baseline value of the outcome and other main baseline measures as covariates. Secondary outcomes will be analysed with linear and generalised linear mixed-effects models. All analyses will adopt the intention to treat principle but will be a complete case analysis (due to the nature of the trial). Covariates to be included in the model are baseline measures of the outcome, other baseline measures, trial arm, stratifiers and any predictor of missingness. The number and proportion of participants with missing data for each baseline, primary and secondary variables will be summarised as a whole and by group and follow-up time point. Descriptive statistics will be used to compare the baseline characteristics of the missing follow-ups (at 1, 3 and 6 weeks) with those of the complete follow-up and, if possible, based on the number of cases, using the logistic predictors of the missing model

A full and detailed Statistical Analysis Plan will be drafted following King's Clinical Trial Unit Standard Operating Procedures and approved by the DMC independent statistician during the study. The Statistical Analysis Plan will be made available via the sponsor's trial webpages.

## Ethics and dissemination

All participants will be required to provide written informed consent. The trial was approved by the National Research Ethics Committee (London, Brent) and the Health Research Authority, reference 20/LO/0206, and was registered with the Medicines and Healthcare Products Regulatory Agency.

The trial was prospectively registered with EUDRACT and retrospectively with clinicaltrials.gov. Protocol modifications will be notified to the National Research Ethics Committee and Health Research Authority and reconsent will be sought from participants, if necessary.

A manuscript with the results of the primary outcome of this study will be published in a peer-reviewed journal. Secondary outcomes may be included in the primary manuscript, or further manuscripts, as appropriate. All manuscripts will be submitted for publication in peer-reviewed journals. The results of the trial may be discussed and publicised via other media.

**Contributors** JR: principal investigator, principal designer of the protocol and wrote the manuscript. HJ: designer of statistical methods and trial monitoring in the protocol and made comments on the manuscript. TM made comments on the protocol and the manuscript. CB made comments on the protocol and the manuscript. NLM: designer of the psychological support model for the protocol and made comments on the manuscript. GK: designer of the psychological support model for the protocol and made comments on the manuscript. FR: oversaw the development of the psychological support model for the protocol and made comments on the manuscript. CD: designed specific scales and processes in the protocol and made comments on the manuscript. BC: senior statistician on the project and made comments on the protocol and the manuscript. AY: chief investigator on the project, made comments on the protocol and the manuscript.

**Funding** The trial is funded by the Clinician Scientist Programme (CS-2017-17-007) from the National Institute for Health Research (NIHR). Psilocybin and placebo capsules are provided without charge by COMPASS Pathways. This work presents independent research part-funded by the National Institute for Health Research (NIHR) Biomedical Research Centre at South London and Maudsley NHS Foundation Trust and King's College London. The views expressed are those of the author(s) and not necessarily those of the NHS, the NIHR or the Department of Health. No award/grant number is applicable.

**Competing interests** JR is an honorary consultant psychiatrist at The South London & Maudsley NHS Foundation Trust, a consultant psychiatrist at Sapphire Medical Clinics and an NIHR Clinician Scientist Fellow at the Centre for Affective Disorders at King's College London. JR's salary is funded by a fellowship (CS-2017-17-007) from the National Institute for Health Research (NIHR). JR leads the Psychedelic Trials Group with Professor Allan Young at King's College London. King's College London receives grant funding from COMPASS Pathways PLC and Beckley PsyTech to undertake phase 1 and phase 2 trials with psychedelics, including psilocybin. COMPASS Pathways PLC has paid for James Rucker to attend trial related meetings and conferences to present the results of research using psilocybin. COMPASS Pathways provided the psilocybin and placebo capsules for this trial, without charge. JR asserts that COMPASS Pathways had no influence over the content of this article or the design of this trial. JR has undertaken paid consultancy work for Beckley PsyTech and Clerkenwell Health. Payments for consultancy work are received and managed by King's College London. James Rucker does not benefit personally. JR has no shareholdings in pharmaceutical companies. Allan H Young. Employed by King's College London; Honorary Consultant SLaM (NHS UK). Paid lectures and advisory boards for the following companies with drugs used in affective and related disorders: Astrazenaca, Eli Lilly, Lundbeck, Sunovion, Servier, Livanova, Janssen, Allegan, Bionomics, Sumitomo Dainippon Pharma, COMPASS. Consultant to Johnson & Johnson. Consultant to Livanova. Received honoraria for attending advisory boards and presenting talks at meetings organised by LivaNova. Principal Investigator in the Restore-Life VNS registry study funded by LivaNova. Principal Investigator on ESKETINTRD3004: 'An Open-label, Long-term, Safety and Efficacy Study of Intranasal Esketamine in Treatment-resistant Depression'. Principal Investigator on 'The Effects of Psilocybin on Cognitive Function in Healthy Participants'. Principal Investigator on 'The Safety and Efficacy of Psilocybin in Participants with Treatment-Resistant Depression (P-TRD)'. Grant funding (past and present): NIMH (USA); CIHR (Canada); NARSAD (USA); Stanley Medical Research Institute (USA); MRC (UK); Wellcome Trust (UK); Royal College of Physicians (Edin); BMA (UK); UBC-VGH Foundation (Canada); WEDC (Canada); CCS Depression Research Fund (Canada); MSFHR (Canada); NIHR (UK). Janssen (UK). No shareholdings in pharmaceutical companies. All other authors declare no competing interests.

**Patient consent for publication** Not applicable.

**Provenance and peer review** Not commissioned; externally peer-reviewed.

**ORCID iDs**
James Rucker http://orcid.org/0000-0003-4647-8088
Ben Carter http://orcid.org/0000-0003-0318-8865

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
