## [Reviewer comments · BMJ Open]

ARTICLE DETAILS

TITLE (PROVISIONAL)	Psilocybin-assisted therapy for the treatment of resistant major depressive disorder (PsiDeR): protocol for a randomised, placebo-controlled feasibility trial
AUTHORS	Rucker, James; Jafari, Hassan; Mantingh, Tim; Bird, Catherine; Modlin, Nadav; Knight, Gemma; Reinholdt, Frederick; Day, Camilla; Carter, Ben; Young, Allan

VERSION 1 – REVIEW

REVIEWER	Fogelson, David UCLA, Psychiatry and Biobehavioral Sciences, Semel Institute for Neuroscience and Human Behavior
REVIEW RETURNED	07-Sep-2021

GENERAL COMMENTS	1. The raters are appropriately blinded. The subjects, while blinded to which treatment they are receiving, active or placebo, can easily guess which treatment they have received. Therefore this study more closely resembles a single blind study and should make some reference to this in the study limitations. It would be good to have the subjects guess which treatment they have received to test for blindness.2. The authors indicate that subjects may be stratified by neuroimaging and "omics". Stratification by presence or absence of a personality disorder (clinical phenotype) might prove equally informative about who responds. Personality disorder might be measured by a rating scale or by a DSM 5 diagnosis. I would not exclude patients with a personality disorder from this study as they comprise a majority of treatment resistant depressed patients in clinical practice. You cannot say, as you do, that a "Relatively wide recruitment criteria allow a more 'real world' sample of participants to be collected" if you exclude patients with a personality disorder.3. Given that this is a phase 2 feasibility trial, measuring personality disorder is not a requirement but would be informative in understanding the study population. Similarly gathering a history of childhood or adult trauma would be informative.4. "Except for esketamine, there have been no significant medical treatment developments for MDD since the introduction of the selective serotonin reuptake inhibitors (SSRIs) in the late 1980s." This is somewhat inaccurate. It would be more accurate to say there have been no significant new antidepressant medication treatment developments.... Other medical treatments for MDD have been developed since the late 1980's, including several atypical antipsychotic medications (aripiprazole, olanzapine, and others), deep brain stimulation, and transcranial magnetic stimulation.5. Would it be possible to have an "active" placebo to create a better double blind?
---

	6. The overall protocol is well designed but I cannot find that a structured clinical interview is performed to screen for major Axis 1 disorders. The protocol says DSM criteria will be used for MDD but does not say how other disorders will be measured as present or absent. If you are not already doing so, using a structured intake interview such as the SCID, structured clinical interview for DSM disorders is a necessity. Similarly there should be some structured interview or rating scale to screen for personality disorder.s
--	--

REVIEWER	Majic, Tomislav Charité Universitätsmedizin Berlin, Department of Psychiatry and Psychotherapy
REVIEW RETURNED	25-Sep-2021

GENERAL COMMENTS	The present manuscript provides a description of the design of a study investigating safety and efficacy of psilocybin in treatment resistant major depression. The design is presented consistently and sound. There are a few minor issues that would have to be addressed before acceptance can be recommended: Page 4: ABSTRACT: ‘package of psychological therapy’: are there specific psychotherapeutic concepts that will be used? Maybe this information could also be added at some other point of the manuscript Page 5: LIMITATIONS: another limitation that the authors might want to add is single dosing, possibly leading to additionally worse outcome in the placebo group due to expectation bias, i.e. deception of patients. INTRODUCTION: Generally, it would be helpful for the reader to add one sentence informing about the characteristics of psychedelic drug effects which in contrast to most other substances are not used regularly, but unfold effects that by far outlast acute drug effects. TRD criteria: would two different antidepressants also include two different SSRIs or are substances with different MOAs required? Line 5: the authors might want to say “characteristic subjective effects” instead of using “effect”, given the broad variety of psychedelic drug effects. Page 6: TRIAL OBJECTIVES: (see also page 10) - I haven’t found information on the nature of the placebo. The authors might want to add inactive placebo or describe the placebo substance used. It would also be interesting to learn about the choice for that specific placebo (and why an inactive placebo appears to have been chosen).
---

	Page 7: PATIENT & PUBLIC INVOLVEMENT: I really like the idea of engaging patients from other trials in the design of the trial, great! Page 9: PREPARATION: Psychological interventions are not described. I'm aware that this is not the main focus of the manuscript, but it would be interesting to know if there are any specific interventions planned, if there is a specific therapeutical background or manual etc. used by psychologists. DOSING: The authors mention that relaxing music will be used. It would be interesting to learn if there is a fixed playlist for all patients or if playlists are varied and how music is selected. Page 11: FOLLOW UP: Integration: Given that the term integration is broadly used in psychedelic therapy science, but definition and data are rather scarce, the authors might want to provide a very short outline of techniques that will be used. Are the 4 hours of psychological support post dosing mandatory for all patients, are these sessions done one on one, and are they included in the fixed sessions mentioned above (at day 1, 1 week, etc.), or is there flexibility in the order of sessions? PARTICIPANT RETENTION Participants will receive ongoing psychological therapy – by whom? Their own psychotherapists or the study team? Page 12: CLINICAL OUTCOMES: Are there any specific hypotheses that will be tested by GAD-7, EQ-5D-5L and WSAS? Page 13: OPEN LABEL EXTENSION: Does the open label extension also include a placebo condition or are the participants guaranteed to receive psilocybin here? This would be important to know, as it might significantly reduce worsening of outcome in participants in the placebo condition in the original trial due to disappointment PSYCHOLOGICAL THERAPIST TRAINING:
--	---

	Again, it might helpful to learn at least a minimum about psychotherapeutic background of the techniques that are used in the trial.
--	--

VERSION 1 – AUTHOR RESPONSE

Reviewer: 1

Dr. David Fogelson, UCLA

Comments to the Author:

1. The raters are appropriately blinded. The subjects, while blinded to which treatment they are receiving, active or placebo, can easily guess which treatment they have received. Therefore this study more closely resembles a single blind study and should make some reference to this in the study limitations. It would be good to have the subjects guess which treatment they have received to test for blindness.

Thank you. I have added an additional limitation to reflect this, however this may face editorial sanction as it takes the number of strengths and limitations over the journal's limit.

Participants are asked the extent to which they believe they received placebo or psilocybin at follow up, collected electronically so that the data will not be visible to the study team.

2. The authors indicate that subjects may be stratified by neuroimaging and "omics". Stratification by presence or absence of a personality disorder (clinical phenotype) might prove equally informative about who responds. Personality disorder might be measured by a rating scale or by a DSM 5 diagnosis. I would not exclude patients with a personality disorder from this study as they comprise a majority of treatment resistant depressed patients in clinical practice. You cannot say, as you do, that a "Relatively wide recruitment criteria allow a more 'real world' sample of participants to be collected" if you exclude patients with a personality disorder.

Thank you. We do not stratify participants by omics or neuroimaging specifically, rather by age group, sex and past psilocybin use. The recruitment criteria being wide is in reference to the degree of treatment resistance. Whilst I appreciate the point re personality disorders, the resource constraints and the nature of the drug, in combination, do not make an ethical case for the inclusion of people with these sorts of difficulties. They often bring complex forms of relational trauma that psilocybin tends to amplify. It isn't feasible for the study team to contain and work with this safely and risks the participants feeling unsupported during the trial and rejected/abandoned after the trial. This has happened in other work we have done. It would need a much better resourced trial than this one, and this is generally a level of resource that no funder, thus far, has shown interest in providing.

3. Given that this is a phase 2 feasibility trial, measuring personality disorder is not a requirement but would be informative in understanding the study population. Similarly gathering a history of childhood or adult trauma would be informative.

Thank you. We measure personality disorder clinically, and measure personality traits (OCEAN) via the TSDI scale. Similarly, we collect information on childhood and adult trauma by the CTS scale. This is discussed in the supplementary information.

4. "Except for esketamine, there have been no significant medical treatment developments for MDD since the introduction of the selective serotonin reuptake inhibitors (SSRIs) in the late 1980s." This is somewhat inaccurate. It would be more accurate to say there have been no significant new antidepressant medication treatment developments.... Other medical treatments for MDD have been

developed since the late 1980's, including several atypical antipsychotic medications (aripiprazole, olanzapine, and others), deep brain stimulation, and transcranial magnetic stimulation.

Thank you. I have amended this.

5. Would it be possible to have an "active" placebo to create a better double blind?

Thank you. We considered the point at initial discussions with regulators and funders. We were advised not to have an active control because they felt that a clean comparison of Adverse Event rates between psilocybin and a true placebo would be informative at this stage, and given the design of previous trials. It is not realistic to change this now we have started recruitment and dosing sessions.

6. The overall protocol is well designed but I cannot find that a structured clinical interview is performed to screen for major Axis 1 disorders. The protocol says DSM criteria will be used for MDD but does not say how other disorders will be measured as present or absent. If you are not already doing so, using a structured intake interview such as the SCID, structured clinical interview for DSM disorders is a necessity. Similarly there should be some structured interview or rating scale to screen for personality disorder.

Thank you. We use the MINI v7, which is a structured clinical interview for DSM5. This is stated in the main manuscript and discussed further in the supplementary information.

Reviewer: 2

Dr. Tomislav Majic, Charité Universitätsmedizin Berlin, Psychiatrische Universitätsklinik der Charité im St. Hedwig-Krankenhaus

Comments to the Author:

The present manuscript provides a description of the design of a study investigating safety and efficacy of psilocybin in treatment resistant major depression. The design is presented consistently and sound. There are a few minor issues that would have to be addressed before acceptance can be recommended:

Page 4:

ABSTRACT:

‘package of psychological therapy’: are there specific psychotherapeutic concepts that will be used? Maybe this information could also be added at some other point of the manuscript

The description of the psychological support model is included in Section 5.1.4.7 of the Supplementary Information. I think it is fair to say that such models are still actively under development, so we cannot be precise here (and a certain degree of flexibility is, in any event, needed for this paradigm).

Page 5:

LIMITATIONS:

another limitation that the authors might want to add is single dosing, possibly leading to additionally worse outcome in the placebo group due to expectation bias, i.e. deception of patients.

Thank you. The journal only allows 5 strengths and limitations. I have added your suggestion (well made) but it may face editorial sanction. Part of the point of the open label extension (in which everyone gets 25mg of psilocybin) was also to mitigate to an extent against this form of bias. I think it is not possible to account for it entirely.

INTRODUCTION:

Generally, it would be helpful for the reader to add one sentence informing about the characteristics of psychedelic drug effects which in contrast to most other substances are not used regularly, but unfold effects that by far outlast acute drug effects.

Thank you. I have added this.

TRD criteria: would two different antidepressants also include two different SSRIs or are substances with different MOAs required?

Two SSRIs would be sufficient.

Line 5: the authors might want to say "characteristic subjective effects" instead of using "effect", given the broad variety of psychedelic drug effects.

Thank you. I have amended this.

Page 6:

TRIAL OBJECTIVES:

(see also page 10) - I haven't found information on the nature of the placebo. The authors might want to add 'inactive' placebo or describe the placebo substance used. It would also be interesting to learn about the choice for that specific placebo (and why an inactive placebo appears to have been chosen).

Thank you. I have amended this. We initially considered an active placebo. This was modified to an inactive placebo after discussion with regulators, who suggested that a comparison of Adverse Event rates between the psilocybin and a true placebo arm would be more useful in this trial, given the design of previous trials. The placebo is Starch 1500, matched for fill volume with the active capsules.

Page 7:

PATIENT & PUBLIC INVOLVEMENT:

I really like the idea of engaging patients from other trials in the design of the trial, great!

Thank you.

Page 9:

PREPARATION:

Psychological interventions are not described. I'm aware that this is not the main focus of the manuscript, but it would be interesting to know if there are any specific interventions planned, if there is a specific therapeutical background or manual etc. used by psychologists.

Thank you. It is described in the supplementary information.

DOSING:

The authors mention that relaxing music will be used. It would be interesting to learn if there is a fixed playlist for all patients or if playlists are varied and how music is selected.

We have a variety of playlists that participants can choose from, or they can bring their own. Generally, we find participants choose one of our playlists. There has been much debate over whether we should be proscriptive about the music that we offer. The general consensus is that it is probably better to give participants a sense of agency here, particularly given that the mental health problems they bring often leave them feeling without agency with respect to their inner world.

Page 11:

FOLLOW UP:

Integration: Given that the term integration is broadly used in psychedelic therapy science, but definition and data are rather scarce, the authors might want to provide a very short outline of techniques that will be used. Are the 4 hours of psychological support post dosing mandatory for all patients, are these sessions done one on one, and are they included in the fixed sessions mentioned above (at day 1, 1 week, etc.), or is there flexibility in the order of sessions?

Thank you. This is covered in the supplementary information. We set minimum amounts of time for psychological preparation and integration however there is the facility to provide additional support if clinically necessary. We keep a record of the total number of hours given at each stage.

PARTICIPANT RETENTION

Participants will receive ongoing psychological therapy – by whom? Their own psychotherapists or the study team?

Thank you. Participants have a therapist who accompanies them through the trial, provided by the study team. This is covered in the supplementary information. I have amended this slightly to make it clear that therapeutic consistency is the goal (although not always logistically feasible in a trial with a long follow up period)

Page 12:

CLINICAL OUTCOMES:

Are there any specific hypotheses that will be tested by GAD-7, EQ-5D-5L and WSAS?

Thank you. Not hypotheses as such, however the exploratory outcomes pertaining to these scales are described in the supplementary information.

Page 13:

OPEN LABEL EXTENSION:

Does the open label extension also include a placebo condition or are the participants guaranteed to receive psilocybin here? This would be important to know, as it might significantly reduce worsening of outcome in participants in the placebo condition in the original trial due to disappointment

Thank you. Everyone is offered 25mg of psilocybin in the open label extension and part of the point is to mitigate against expectation bias.

PSYCHOLOGICAL THERAPIST TRAINING:

Again, it might helpful to learn at least a minimum about psychotherapeutic background of the techniques that are used in the trial.

Thank you. It is described in the supplementary information.

*** **

COI statements:

Reviewer: 1

Competing interests of Reviewer: None

Reviewer: 2

Competing interests of Reviewer: None

VERSION 2 – REVIEW

REVIEWER	Fogelson, David UCLA, Psychiatry and Biobehavioral Sciences, Semel Institute for Neuroscience and Human Behavior
REVIEW RETURNED	15-Oct-2021

GENERAL COMMENTS	1. Are you planning an extension to this study beyond the 6 weeks follow up? It would contribute greatly to our understanding of the durability of psilocybin treatment if there were follow-ups at 3, 6, and 12 months.2. I would change the last sentence of the first paragraph of the introduction to read: "Except for esketamine, there have been no significant medication developments..." Otherwise you are implying trans cranial magnetic stimulation has not been a significant development.3. I would recommend a score of at least 17 on the Hamilton Depression Rating Scale. There is consensus that the following severity ranges for the HAMD are accurate: no depression (0-7); mild depression (8-16); moderate depression (17-23); and severe depression (≥ 24). Why require at least "moderate" depression. Scores less than 17 are likely to include patients with dysthymia, not Major Depressive Disorder who are less likely to respond to medication treatments.4. As I have stated before, I do not understand why you would exclude all patients with a DSM-5 personality disorder. You are going to have no patients left for your study with this exclusion.5. Blinding continues to be a problem for your study as patients can easily guess whether or not they received psilocybin. I recommend an active placebo
---

REVIEWER	Majic, Tomislav Charité Universitätsmedizin Berlin, Department of Psychiatry and Psychotherapy
REVIEW RETURNED	10-Oct-2021
GENERAL COMMENTS	No more comments. Acceptance is recommended

VERSION 2 – AUTHOR RESPONSE

Reviewer: 1

Dr. David Fogelson, UCLA

Comments to the Author:

1. Are you planning an extension to this study beyond the 6 weeks follow up? It would contribute greatly to our understanding of the durability of psilocybin treatment if there were follow-ups at 3, 6, and 12 months.

Thank you. The paper already describes an open label extension to the study that extends 3 months beyond 6 week follow up. We are not funded or ethically authorised for follow up at 6 or 12 months, however we will certainly consider this for future studies.

2. I would change the last sentence of the first paragraph of the introduction to read: "Except for esketamine, there have been no significant medication developments..." Otherwise you are implying trans cranial magnetic stimulation has not been a significant development.

Thank you. I have added trans cranial magnetic stimulation to the sentence.

3. I would recommend a score of at least 17 on the Hamilton Depression Rating Scale. There is consensus that the following severity ranges for the HAMD are accurate: no depression (0-7); mild depression (8-16); moderate depression (17-23); and severe depression (≥ 24). Why require at least "moderate" depression. Scores less than 17 are likely to include patients with dysthymia, not Major Depressive Disorder who are less likely to respond to medication treatments.

Thank you. The cut off score of 14 takes account of partial but inadequate response to antidepressants. I do appreciate the comment and we will consider the suggestion in a future amendment to the trial protocol and for future protocols. Psilocybin assisted therapy is very unlikely ever to be a treatment for severe depression.

4. As I have stated before, I do not understand why you would exclude all patients with a DSM-5 personality disorder. You are going to have no patients left for your study with this exclusion.

We are recruiting adequately with the inclusion criteria as they are. It is not practical to support people with complex mental health problems like personality disorder in a trial of psilocybin assisted therapy like this. Hence the exclusion.

5. Blinding continues to be a problem for your study as patients can easily guess whether or not they received psilocybin. I recommend an active placebo

Thank you. Please see recent Nature Medicine commentary for a discussion about this issue.
<https://www.nature.com/articles/s41591-021-01524-1>

The trial is already underway, thus not practical to remanufacture an active placebo now.

We were requested by the funder to perform a true placebo controlled study.

I have added this to the manuscript.

Reviewer: 2

Dr. Tomislav Majic, Charité Universitätsmedizin Berlin, Psychiatrische Universitätsklinik der Charité im St. Hedwig-Krankenhaus

Comments to the Author:

No more comments. Acceptance is recommended